# Concomitant Nrf2- and ATF4-Activation by Carnosic Acid Cooperatively Induces Expression of Cytoprotective Genes

**DOI:** 10.3390/ijms20071706

**Published:** 2019-04-05

**Authors:** Junsei Mimura, Atsushi Inose-Maruyama, Shusuke Taniuchi, Kunio Kosaka, Hidemi Yoshida, Hiromi Yamazaki, Shuya Kasai, Nobuhiko Harada, Randal J. Kaufman, Seiichi Oyadomari, Ken Itoh

**Affiliations:** 1Department of Stress Response Science, Center for Advanced Medical Research, Hirosaki University Graduate School of Medicine, Hirosaki 036-8562, Japan; ainose@tohoku-mpu.ac.jp (A.I.-M.); hiromi-y@hirosaki-u.ac.jp (H.Y.); kasai-s@hirosaki-u.ac.jp (S.K.); nharada@med.tohoku.ac.jp (N.H.); itohk@hirosaki-u.ac.jp (K.I.); 2Department of Microbiology, Tohoku Medical and Pharmaceutical University, Sendai 981-8558, Japan; 3Division of Molecular Biology, Institute of Advanced Medical Sciences, The University of Tokushima, Tokushima 770-8503, Japan; taniuchi@genome.tokushima-u.ac.jp (S.T.); oyadomar@genome.tokushima-u.ac.jp (S.O.); 4Research and Development Center, Nagase & Co. Ltd., Kobe 651-2241, Japan; kunio.kosaka@nagase.co.jp; 5Department of Vascular Biology, Institute of Brain Science, Hirosaki University Graduate School of Medicine, Hirosaki 036-8562, Japan; hidemi_yoshida@gakushikai.jp; 6Institute for Animal Experimentation, Tohoku University Graduate School of Medicine, Sendai 980-8575, Japan; 7Degenerative Diseases Research Program, Sanford Burnham Prebys Medical Discovery Research Institute, La Jolla, CA 92037, USA; rkaufman@sbpdiscovery.org

**Keywords:** Nrf2, ATF4, NGF, carnosic acid, antioxidant

## Abstract

Carnosic acid (CA) is a phytochemical found in some dietary herbs, such as *Rosmarinus officinalis* L., and possesses antioxidative and anti-microbial properties. We previously demonstrated that CA functions as an activator of nuclear factor, erythroid 2 (NF-E2)-related factor 2 (Nrf2), an oxidative stress-responsive transcription factor in human and rodent cells. CA enhances the expression of nerve growth factor (NGF) and antioxidant genes, such as *HO-1* in an Nrf2-dependent manner in U373MG human astrocytoma cells. However, CA also induces *NGF* gene expression in an Nrf2-independent manner, since 50 μM of CA administration showed striking *NGF* gene induction compared with the classical Nrf2 inducer *tert*-butylhydroquinone (tBHQ) in U373MG cells. By comparative transcriptome analysis, we found that CA activates activating transcription factor 4 (ATF4) in addition to Nrf2 at high doses. CA activated ATF4 in phospho-eIF2α- and heme-regulated inhibitor kinase (HRI)-dependent manners, indicating that CA activates ATF4 through the integrated stress response (ISR) pathway. Furthermore, CA activated Nrf2 and ATF4 cooperatively enhanced the expression of *NGF* and many antioxidant genes while acting independently to certain client genes. Taken together, these results represent a novel mechanism of CA-mediated gene regulation evoked by Nrf2 and ATF4 cooperation.

## 1. Introduction

Redox homeostasis is essential for normal cellular function and viability. NF-E2-related factor 2 (Nrf2) is a cap‘n’collar (CNC)-basic leucine zipper (bZip) transcription factor, which plays a central role in maintaining redox homeostasis. Nrf2 transactivates several antioxidant genes, such as heme oxygenase-1 (*HO-1*) and thioredoxin reductase 1 (*TXNRD1*), in response to oxidative stress [1,2,3]. Nrf2 directly modulates the expression of these target genes through binding to its recognition DNA element, the antioxidant responsive element (ARE) (otherwise known as the electrophile response element (EpRE)), as an Nrf2/small Mafs (i.e., MafK, MafF, and MafG) heterodimer [1,2,3]. Nrf2 activity is mainly regulated by Kelch-like erythroid cell-derived protein with CNC homology (ECH)-associated protein 1 (Keap1), which mediates Nrf2 degradation through the ubiquitin–proteasome system (UPS) during non-stressed conditions. Nrf2 is activated upon Keap1 inactivation through oxidation of its redox-sensitive cysteine residues (such as Cys151, Cys273, and Cys288) via various prooxidative reagents, such as reactive electrophiles (e.g., *tert*-butylhydroquinone (tBHQ) and sulforaphane (SFN)), heavy metals, and reactive oxygen/nitrogen species (ROS/RNS) [1,3]. We previously demonstrated that carnosic acid (CA) activates Nrf2 through Keap1 cysteine modification. CA is an abietane-type diterpenoid abundantly found in herbal plants, such as *Rosmarinus officinalis* L. and *Salvia officinalis* [4,5]. CA exhibits antioxidative and anti-microbial activities in these plants [4,5]. We previously found that CA induces nerve growth factor (NGF) in U373MG human astrocytoma cells, T98G human glioma cells, and normal human astrocytes in an Nrf2-dependent manner [6,7] and that CA induces neurite outgrowth of PC12h cells through an Nrf2- and p62-dependent pathway [8]. Neuroprotective effects of CA in animals were demonstrated in middle cerebral artery occlusion and transgenic Alzheimer’s model mice [9,10]. Our previous results demonstrated that CA preferentially accumulates in astrocytes, in which Nrf2 primarily regulates neuroprotective function in the central nervous system [9,11].

In addition to Nrf2, several transcription factors, such as forkhead box, class O family (FOXOs), c-jun/activator protein 1 (AP-1), and activating transcription factor 4 (ATF4), contribute to redox homeostasis by inducing antioxidant genes [12,13]. Among them, c-jun and ATF4 were shown to dimerize with Nrf2, suggesting cooperation between Nrf2 and c-jun or ATF4 [14,15]. AP-1 binding sequences (TRE) are often embedded in the AREs of many antioxidant genes and regulated by AP-1. In addition, ATF4 binds to different cis-element named amino acid response elements (AAREs) [12,16,17]. ATF4 is mainly activated via a unique stress response pathway named integrated stress response (ISR) [12,16,17,18]. In mammalian ISR, four eIF2α kinases, (heme regulated inhibitor of translation (HRI), dsRNA-activated protein kinase (PKR), PKR-like endoplasmic reticulum kinase (PERK) and general control nonderepressible-2 (GCN2)) catalyze phosphorylation of eukaryotic initiation factor 2α (eIF2α) at serine51 residue in response to different stress signals; heme deficiency (HRI), viral infection (PKR), ER stress (PERK), and amino acid deficiency (GCN2) [18]. Upon eIF2α phosphorylation, the preinitiation complex formation is prevented, and thereby general translation is suppressed, but translation from mRNA containing specific 5′-UTR structure (i.e., upstream open reading frame (uORF) or internal ribosome entry site (IRES)), such as *ATF4* or virus mRNA, are rather induced [18]. We previously demonstrated that Nrf2 and ATF4 cooperatively modulate gene expression of cystem x_c_
^−^ transporter-related protein (xCT), a cystine antiporter x_c_
^−^ system, in T24 human bladder cancer cells. The *xCT* gene regulatory region has both ARE and AAREs, and Nrf2 and ATF4 cooperatively modulate *xCT* gene expression by interacting with each other within the gene regulatory regions [19]. However, the significance of Nrf2 and ATF4 cooperation remains largely unclear.

Here we have studied the impact of ATF4 on CA-inducible gene expression in U373MG cells. We found that 50 μM of CA activate ATF4 in addition to Nrf2, and these two transcription factors cooperatively activate CA induction of antioxidant genes and *NGF*. These results provide evidence of a novel mechanism of CA action, which is mediated by Nrf2 and ATF4 collaboration.

## 2. Results

### 2.1. Dose-Dependent Induction of NGF and AKR1B10 by CA

In previous work, we demonstrated that CA induces *NGF* gene expression in an Nrf2-dependent manner in T98G and U373MG human glioblastoma cells [6,7]. A subsequent dose-response analysis revealed that CA remarkably induced *NGF* in U373MG cells at a high-dose (50 μM) (Figure 1A), which is reminiscent of our previous observation in T98G glioblastoma cells [20]. *NGF* was only marginally induced by 10 or 20 μM of CA (1.3- or 1.5-fold, respectively), but strongly induced by 50 μM of CA (19.7-fold) (Figure 1A). On the other hand, aldo-keto reductase 1B10 (*AKR1B10*), which is known as a typical Nrf2/ARE gene, was induced 13.7-fold by 10 μM CA and 26.9-fold at 50 μM of CA (Figure 1B). Curiously, *tert*-butylhydroquinone (tBHQ, a classical Nrf2 inducer), only slightly affected *NGF* gene expression even at a high-dose (2.0-fold at 50 μM of tBHQ), but significantly induced *AKR1B10* to relatively the same extent as CA (15.8-fold at 50 μM of tBHQ) (Figure 1A,B). Surprisingly, CA and tBHQ similarly induced Nrf2 protein accumulation, although they resulted in varying *NGF* induction (Figure 1A,C). Furthermore, Nrf2 knockdown completely abolished CA induction of *AKR1B10*, but only partially decreased that of *NGF*, although Nrf2 accumulation was below the level of detection (Figure 1D–F). Taken together, these results suggest that the gene regulatory mechanism of *NGF* is distinct from that of *AKR1B10* and that the *NGF* gene is regulated not only by Nrf2 but also by unknown factor or factors.

### 2.2. High-Dose CA Activates the ATF4 Pathway

To investigate the molecular mechanism involved in high-dose CA induction of *NGF*, we performed a transcriptome analysis of U373MG cells treated with DMSO (vehicle) or CA (50 μM). Transcriptome data were obtained by a conventional DNA chip-based microarray analysis as described in Section 4.3. As expected, microarray analysis showed upregulation of the expression of several Nrf2 target genes in CA-treated samples, such as *AKR1B10*, *HO-1*/*HMOX1*, and *TXNRD1* (18.8-, 12.0-, and 4.6-fold, respectively) (Figure 2A, Appendix A). Although CA reportedly functions as an agonist of peroxisome proliferator-activated receptor γ (PPARγ) [21], PPARγ pathway gene inductions were not observed in U373MG cells (data not shown). Notably, with CA treatment, we found upregulation of *ATF4* (2.6-fold) and ATF4/AARE genes, such as asparagine synthetase (*ASNS*) and tribbles homolog 3 (*Trib3*) (4.1- and 2.5-fold, respectively) (Figure 2A, Appendix A). To confirm the microarray result, we next performed RT-qPCR and immunoblot analysis. Curiously, the induction of ATF4 mRNA and protein was specifically observed by treatment with 50 μM, but not 10 or 20 μM of CA, whereas Nrf2 protein induction was detectable following addition of 10 μM of CA and increased until 50 μM of CA (Figure 2B,C). Similar to ATF4 expression, the ATF4 target genes *ASNS*, *Trib3*, and C/EBP homologous protein (*CHOP*), which are regulated in an AARE-dependent manner, were significantly induced by 50 μM, but not 10 or 20 μM of CA (Figure 2D). Inhibition of transcription or translation (by actinomycin D (ActD) or cycloheximide (CHX), respectively) abolished CA induction of ATF4 protein as well as mRNA (Figure 2E,F), suggesting the requirement of *de novo* mRNA and protein synthesis for CA induction of ATF4, although CHX treatment slightly up-regulated *ATF4* mRNA expression even in the absence of CA by unknown reason. ATF4 activation was also confirmed by reporter assay and chromatin immunoprecipitation (ChIP) analysis. The luciferase reporter gene, which contains the AARE of the human *ASNS* gene (pAARE-SV), as well as the Nrf2-responsive ARE-driven reporter gene (pARE-SV), was significantly upregulated by 50 μM of CA (Figure 2G). CA-inducible ATF4 recruitment to the AARE within the human *ASNS* gene promoter was also observed by ChIP analysis (Figure 2H). ATF4 activation by CA was also observed in other human cancer cell lines, such as HeLa, Caco-2, and HepG2 cells, suggesting that CA induction of ATF4 is not restricted to astrocytoma cells, although they exhibited varying CA dose-dependency (Appendix A). Collectively, these results suggest that high-dose CA activates the ATF4 and Nrf2 pathways. Because Nrf2 functions primarily in astrocytes during protection against neurodegeneration [11], in the present study, we focused on elucidating the role of CA within U373MG human astorcytoma cells.

### 2.3. Nrf2 and ATF4 Cooperatively Contribute to CA-Inducible NGF Gene Expression and Other Antioxidant Genes

We performed RNA interference (RNAi) experiments to investigate the role of Nrf2 and ATF4 in individual CA-inducible genes. Knockdown of Nrf2 and ATF4 was validated by immunoblot and RT-qPCR analyses (Figure 3A–C). Intriguingly, knockdown of either Nrf2 or ATF4 significantly reduced CA induction of the *NGF* gene in U373MG cells (Figure 3D). On the other hand, CA induction of *AKR1B10* was almost totally abolished by Nrf2 knockdown but only slightly by ATF4 knockdown without reaching statistical significance (Figure 3E). CA induction of *ASNS* and other ATF4 target genes, such as *Trib3* and *CHOP*, was abolished by ATF4 knockdown, but not by Nrf2 knockdown (Figure 3F, and Appendix A). These results indicate that *AKR1B10* is regulated mainly by Nrf2, whereas *ASNS* gene expression is regulated exclusively by ATF4.

Of note, similar to *NGF*, several antioxidant genes of the Nrf2/ARE pathway, such as *HO-1*, *TXNRD1*, *xCT*, glutamate-cysteine ligase catalytic subunit (*GCLC*), and glutamate-cysteine ligase modifier subunit (*GCLM*), were also significantly increased by 50 μM of CA versus the same dose of tBHQ (Appendix A). Consistently, induction of these genes by 50 μM of CA was reduced by either Nrf2 or ATF4 knockdown and severely diminished by Nrf2 and ATF4 double knockdown (Figure 3G,H, and Appendix A). Conversely, *NQO1* was similarly induced by 50 μM of CA as well as 50 μM of tBHQ, while ATF4 knockdown did not significantly affect CA induction of *NQO1* gene expression (Appendix A), suggesting that ATF4 affects Nrf2/ARE gene expression in a gene-specific manner. Nrf2 and ATF4 cooperation was also confirmed by Nrf2 and ATF4 coactivation by tBHQ and tunicamycin (Tm). Of note, tBHQ activated Nrf2, but failed to activate ATF4, which may correlate with poor *NGF* inducibility (Figure 4A,B). Tm is an endoplasmic reticulum (ER) stressor that activates ATF4 through the ER stress/PERK pathway. Although ER stress was previously shown to activate Nrf2 through PERK [22], Tm did not activate Nrf2 in U373MG cells at a dose of 2 μg/mL (Figure 4A). In concordance with knockdown experiments, tBHQ and Tm cotreatment highly induced *NGF* compared with tBHQ or Tm treatment alone (Figure 4B). On the other hand, tBHQ/Tm cotreatment rather decreased *AKR1B10* expression compared with tBHQ alone (Figure 4C). In contrast, Tm-induced *ASNS* expression was unaffected by tBHQ addition (Figure 4D). These results indicate that Nrf2 and ATF4 coactivation enhances *NGF* gene expression, while ATF4 activation alone is sufficient to induce *NGF*. As expected, tBHQ and Tm cotreatment also enhanced the expression of antioxidant genes, including *HO-1* and *TXNRD1*, when compared with tBHQ or Tm treatment alone (Figure 4E,F). A cooperative effect of tBHQ and Tm on *NGF* and *HO-1* expression was also observed in normal human astrocytes (NHAs) (Appendix A), suggesting that ATF4 and Nrf2 cooperation is not limited to cancer cells. These results indicate that marked *NGF* induction by high-dose CA is due to Nrf2 and ATF4 coactivation.

### 2.4. CA Activates ATF4 through the ISR Pathway

The integrated stress response (ISR) plays a major role in ATF4 activation following stress [12,16]. In an ISR system, distinct stress signals activate one of eIF2α kinases and are coupled to the phosphorylation of eIF2α at serine 51, and phospho-eIF2α suppresses global translation, but enhances ATF4 translation, by modulating eukaryotic initiation factor 2B (eIF2B) activity [12,16]. To investigate the role of eIF2α phosphorylation during CA induction of ATF4, we performed an immunoblot analysis using an anti-phospho-eIF2α antibody. A time-course analysis revealed that 50 μM of CA induced eIF2α phosphorylation with a peak at 2 h, and subsequent ATF4 induction was observed in U373MG cells (Figure 5A). Inhibition of protein synthesis was also observed with 50 μM of CA (Figure 5B), suggesting that CA suppresses global translation through eIF2α phosphorylation. Furthermore, ISRIB (an ISR inhibitor), which reverses the effect of eIF2α phosphorylation by activating eIF2B [23], abolished CA induction of ATF4, but not Nrf2 (Figure 5C). CA induction of the ATF4 protein, *ATF4*, and its target *ASNS* genes was diminished in mutant mouse embryonic fibroblasts (MEFs), which harbor the non-phosphorylatable eIF2α mutant allele (MEF(A/A)) [24] (Appendix A). Collectively, these results suggest that CA enhances eIF2α phosphorylation to induce ATF4.

### 2.5. CA Activates ISR in An HRI-Dependent Manner

In mammals, four eIF2α kinases (HRI, PKR, PERK, and GCN2) phosphorylate eIF2α in response to various stresses [16,25]. To elucidate which eIF2α kinase may be responsible for CA induction of ATF4, we performed a knockdown experiment by using RNAi (Figure 6A,B). The knockdown efficiency of eIF2α kinases was confirmed by immunoblot analysis (Appendix A). Among these four eIF2α kinase knockdowns, only HRI knockdown completely abolished CA induction of the ATF4 protein and mRNA, as well as *ASNS* mRNA (Figure 6A,B, Appendix A). Previously, Min et al. reported that CA (at 40 μM) activates ATF4 through ER stress induction in human renal carcinoma Caki cells [26]. However, knockdown of PERK, which is responsible for ER stress-mediated ATF4 activation, did not affect CA induction of ATF4 (Figure 6A,B). In addition, CA did not induce ER stress-dependent splicing of X-box binding protein 1 (*XBP1*) mRNA and heat shock protein family A member 5 (*HSPA5*)/binding immunoglobulin protein (*BiP*) gene expression (Appendix A), suggesting that CA is unlikely to induce ATF4 through ER stress in U373MG cells. Together, these results indicate that CA activates ATF4 through the HRI/phospho-eIF2α pathway.

## 3. Discussion

Our studies on CA-inducible gene expression revealed the significance of ATF4-Nrf2 cooperation during the strong expression of *NGF* and other antioxidant genes in human cells. CA activated Nrf2 by Keap1 inactivation at a low (<20 μM) to high-dose (50 μM), but activated ATF4 at a high-dose (50 μM) via HRI- and p-eIF2α-dependent mechanisms (Figure 7). It is likely that low-dose CA is sufficient for the *S*-alkylation of Keap1, while high-dose CA (50 μM) is required for HRI activation. HRI-dependent eIF2α phosphorylation induces ATF4 protein as well as *ATF4* mRNA (Figure 6A,B, Appendix A). The evidence that CA targets Keap1 and HRI in a concentration-dependent manner suggests that CA functions as a multi-target compound like resveratrol. Resveratrol targets multiple proteins with different IC_50_ values and exhibits diverse biological effects [27]. It is possible that CA targets other proteins except for Keap1 and HRI, since CA exhibits various biological effects in addition to neuroprotective effect [28,29]. Further study is required for the elucidation of other CA target proteins.

The knockdown study clearly showed that 50 μM of CA-inducible genes can be categorized into three groups: genes primarily regulated by Nrf2 (*AKR1B10*, *NQO1*), by ATF4 (*ASNS*, *Trib3*), or by both Nrf2 and ATF4 (*NGF*, *HO-1*, etc.) (Figure 3 and Figure 7, Appendix A). In general, Nrf2 and ATF4 mediate different stress responses [1,2,16,27]. Nrf2 is activated mainly by electrophiles or reactive oxygen species (ROS), whereas ATF4 is activated by divergent stresses, including heme deficiency, viral infection, ER stress, and amino acid deficiency [1,2,16,27]. Nrf2 transactivated gene expression through binding as an Nrf2/small Maf (sMaf) heterodimer to the *cis*-element ARE (consensus sequence of TGAC/GNNNGC, with N representing any base) [1,2,3], whereas ATF4 transactivates through AARE (consensus sequence of TGATGCAAT) as an ATF4/CCAAT/enhancer binding protein β (C/EBPβ) heterodimer or ATF4/C/EBPγ heterodimer, as recently reported [16,30,31]. AREs have been found in many antioxidant genes and phase II drug metabolizing genes, as well as proteasomal genes [1,2,3]. On the other hand, AAREs have been found in genes involved in amino acid metabolism, unfolding protein responses, autophagy, and apoptosis [12,16,30]. Recent reports also demonstrated that ATF4 regulates the oxidative stress response by regulating genes involved in transsulfuration pathway and one-carbon (1C) metabolism, promoting the synthesis of glutathione (GSH) and reduced form of nicotinamide-adenine dinucleotide phosphate (NADPH), although the role of AAREs in these genes still remains to be established [32]. Consistently, our microarray analysis also identified the cystathione beta-synthase (*CBS*) and cystathione gamma-lyase (*CTH*) genes of the transsulfuration pathway and the methylenetetrahydrofolate dehydrogenase 2 (*MTHFD2*) gene of 1C metabolism as ATF4 targets, and CA induction of *CBS* and *CTH* exhibited ATF4-dependency (Appendix A, Appendix A).

In the present study, we firstly found that that *NGF* is largely regulated by ATF4 in addition to Nrf2 (Figure 7). By bioinformatics analysis, we found several ARE and AARE consensus sequences in the human *NGF* gene (data not shown); however, it remains unknown whether these elements are functional or not. To the best of our knowledge, only two genes, the cystine antiporter *xCT* and *p62/SQSTM1*, possess both ARE and AARE and are regulated by Nrf2 and ATF4, either independently or cooperatively [19,33,34,35,36]. We confirmed Nrf2- and ATF4-dependent CA induction of *xCT* or *p62* (Appendix A). In addition, we found that CA induction of other antioxidant genes, such as *HO-1*, *TXNRD1*, were also mediated by both Nrf2 and ATF4 (Figure 3G,H). Several studies also suggested ATF4 regulation of these antioxidant genes and its protective effects against oxidative stress [15,33,37]. However, it remains unclear how ATF4 affects the gene expression of *HO-1*, *TXNRD1*, *GCLC*, and *GCLM*. It is possible that these antioxidant genes have also unidentified functional AAREs, but it is of note that an ATF4-dependent, but AARE-independent, *HO-1* gene regulation mechanism has been reported. He et al. reported that ATF4 forms a heterodimer with Nrf2 and modulates *HO-1* gene expression through binding to the stress-responsive element (StRE), which contains AREs, but not AARE [15]. However, it is controversial whether the Nrf2/ATF4 heterodimer directly transactivates *HO-1* through StRE, since contradicting results were also reported, and the affinity of the Nrf2/ATF4 heterodimer to the StRE is significantly lower than that of the Nrf2/sMaf heterodimer [15,38]. Using ChIP analysis, Dey et al. reported that Nrf2 and ATF4 are recruited to the StRE-containing *HO-1* enhancers in a mutually dependent manner and cooperatively activate *HO-1* gene expression upon cancer cells’ detachment from the matrix (i.e., anoikis) [39]. We also showed that ATF4 is recruited to the ARE-containing region in the first intron of the *xCT* gene in an Nrf2-dependent manner upon proteasome inhibition [19]. The nature of this Nrf2-ATF4 interaction on the gene regulatory regions remains to be elucidated.

Several other indirect mechanisms may explain Nrf2-ATF4 cooperation. We previously demonstrated the possibility that ATF4 functions as a coactivator of Nrf2 in *xCT* gene regulation [19]. In addition, ATF4 regulates several genes that affect Nrf2 activity. Both p62 and sestrin-2 (SESN2) activate Nrf2 by degrading Keap1 through an autophagic mechanism [34,40]. Although *p62* and *SESN2* were induced by CA in an ATF4-dependent manner, knockdown of neither gene affected CA induction of *NGF* (data not shown). Curiously, ATF4 knockdown diminished CA induction of *Nrf2* mRNA, suggesting that ATF4 may regulate *Nrf2* gene expression in U373MG cells, although the Nrf2 protein level was largely unaffected by ATF4 knockdown (Figure 3A,C). On the contrary, in retinal pigment epithelial cells, Nrf2 directly transactivates the *ATF4* gene via binding to ARE at the *ATF4* gene promoter [41,42]. However, CA induction of *ATF4* mRNA was scarcely affected by Nrf2 knockdown, suggesting that Nrf2 is not involved in *ATF4* gene regulation within U373MG cells (Figure 3B). Further study is needed in order to elucidate how Nrf2 and ATF4 cooperatively modulate *NGF* and antioxidant gene expression.

At a low-dose, CA activates Nrf2 pathway only and exhibits moderate anti-oxidative effect, while at a high-dose, CA activates both Nrf2 and ATF4 to potentiate Nrf2 antioxidative pathway (Figure 7). Ehren and Maher previously reported that fisetin, a flavonoid, activates both ATF4 and Nrf2 and regulates glutathione homeostasis in the HT22 mouse hippocampal nerve cell line, although which eIF2α kinase is responsible for ATF4 activation by fisetin remains unknown [43]. In U373MG cells, GSH was significantly increased by treatment with 20 μM of CA, but not 50 μM of CA, although both *GCLC* and *GCLM* were significantly induced (data not shown, Appendix A). This paradoxical result could be explained in two ways: (1) a higher concentration of CA depletes cellular GSH by reacting with GSH thiol as suggested previously [9]; or (2) ATF4-mediated induction of cation transporter regulator-like protein 1 (CHAC1), a GSH degrading enzyme, decreases GSH [44]. In fact, CA strongly induced *CHAC1* in an ATF4-dependent manner (Appendix A). Since the overproduction of GSH causes reductive stress, ATF4 may regulate the redox balance by maintaining the GSH level. Inoue et al. also demonstrated 4-hydroperoxy-2-decenoic acid ethyl ester (HPO-DAEE), a synthetic derivative of royal jelly fatty acid, strongly induced *HO-1* by concomitant activation of both Nrf2 and ATF4 [45]. Thus, it would be of great interest to explore natural pharmacological agents that activate Nrf2 and ATF4.

In the present study, we showed that CA activates ATF4 through HRI-dependent pathway. In our preliminary experiments, HRI knockdown also diminished eIF2α phosphorylation (Appendix A) and reversed global protein synthesis inhibition by CA (Appendix A). In addition, CA induction of eIF2α phosphorylation was abolished in the mutant MEFs lacking all four eIF2α kinases (4KO) and restored by reconstitution of the *Hri* gene [46] (Appendix A). Furthermore, CA led to the appearance of a slower-migrating HRI band (Appendix A), which may correspond to the autophosphorylated form of HRI [47,48]. HRI is activated by various stresses in different cell types, such as heme deficiency, heat shock, osmotic, and oxidative stresses [48,49]. It was reported that the HRI/ATF4 pathway plays an important role in the protection from arsenite or cadmium-induced oxidative stress in erythroblasts, suggesting a functional similarity between the Keap1/Nrf2 and HRI/ATF4 pathways [50,51]. HRI has several cysteine residues, which are responsible for heme binding, with heme binding to HRI suppressing its kinase activity. It was reported that cysteine modifications by metal ions, such as Hg^2+^ and nitric oxide (NO), modulate HRI activity [52,53]. Since hemin attenuated the CA induction of ATF4, CA may affect heme binding to HRI (Appendix A). In addition, as CA covalently modifies the cysteine residues of Keap1 [9], we first speculated that CA directly alkylates cysteine residues of HRI. However, HRI was insignificantly pulled down by biotinylated CA (Appendix A), suggesting a much lower affinity of CA for HRI than Keap1, which is consistent with the fact that ATF4 activation requires a higher dose of CA than Nrf2 activation. In addition, it may be difficult to detect CA-HRI interaction by conventional pull-down experiment, since CA may decrease HRI stability by binding to HRI (Appendix A). In contrast, heat shock proteins, such as heat shock protein 70 (HSP70), mediate HRI activation by heat shock and oxidative stress, and the 2-oxoglutarate-dependent oxygenase 2-oxoglutarate and iron-dependent oxygenase domain containing 1 (OGFOD1) is associated with arsenite-induced HRI activation [54,55], suggesting that CA may indirectly activate HRI via these HRI associating factors. Curiously, piciferic acid lacking one hydroxyl group in the catechol ring from CA (redox insensitive) failed to activate both ATF4 and Nrf2 (Appendix A), suggesting that redox regulation may participate in the CA activation of HRI as well as Keap1 inactivation. Further study is required to clarify the molecular mechanism by which CA activates HRI.

In ISR, eIF2α serine 51 phosphorylation plays an important role in the selective translation of ATF4 [16]. CA induction of ATF4 showed eIF2α phosphorylation dependency in U373MG and MEF cells (Figure 5A, Appendix A). However, slight ATF4 induction was still observed in CA-treated MEF(A/A) (Appendix A), indicating that the phospho-eIF2α-independent pathway may be also be involved in CA induction of ATF4 in MEFs. Recent studies suggested that mTORC1 activates ATF4 in a phospho-eIF2α-independent manner in MEFs [56]. Further study is required to elucidate the mechanism by which CA activates ATF4 in MEFs. Although ATF4 activity is mainly regulated by posttranscriptional mechanisms, CA also upregulates *ATF4* gene expression (Figure 2B). ATF4 and nuclear protein 1 (NUPR1) (a downstream gene of ATF4) transactivate *ATF4* gene transcription [57,58]. Since *de novo* protein synthesis is required for CA induction of *ATF4* (Figure 2E,F), positive feedback regulation by ATF4 itself or NUPR1 may be involved in CA induction of *ATF4*.

A number of studies demonstrated that Nrf2 activation by pharmacological agents or phytochemicals is beneficial for oxidative stress-associated disorders, such as neurodegenerative disease [1,3]. As ATF4 may be activated by several cellular stresses, including mitochondrial and ER stress [59], additional pharmacological activation of Nrf2 could strongly induce *NGF* and other antioxidant genes (Figure 1A, Appendix A). Thus, strong *NGF* induction mediated by Nrf2 and ATF4 in astrocytes may be promising for the prevention of Alzheimer’s disease, as the survival of cholinergic neurons that are specifically damaged in the early phase of Alzheimer’s disease is dependent on NGF [60]. Of note, ATF4-induced CHOP is involved in ER and oxidative stress-mediated cell death. It was reported that Nrf2 suppresses *CHOP* gene expression by inhibiting ATF4 binding to AARE at the *CHOP* gene promoter in thyroid cancer cells [61]. Although CA induction of CHOP was not affected by Nrf2 knockdown in U373MG cells (Appendix A), cross talk between Nrf2 and ATF4 may determine cell fate during therapeutic Nrf2 activation within many degenerative diseases.

In summary, our findings provided novel insight into the understanding of the mechanism of CA action, and *NGF* and antioxidant gene regulation mediated by Nrf2 and ATF4.

## 4. Materials and Methods

### 4.1. Materials

Carnosic acid, carnosol, and rosmarinic acid were supplied by Nagase Co LTD. (Kobe, Japan). Piciferic acid was purchased from Tokyo Chemical Industry (Tokyo, Japan). Dimethylsulfoxide, actinomycin D, cycloheximide, tunicamycin, and hemin were obtained from Wako Pure Chemicals (Osaka, Japan). The *tert*-butylhydroquione (tBHQ) was obtained from Kanto Chemical (Tokyo, Japan). ISRIB was obtained from Sigma-Aldrich (St. Louis, MO, USA). Primary antibodies against ATF4 (CREB2) (C-20), Nrf2 (H-300), lamin B (M-20), and HSP90α/β (AC88) were obtained from Santa Cruz Biotechnology (Dallas, TX, USA), and primary antibodies against eIF2α (9722S) and phospho-Ser51 eIF2α (9721S) were purchased from Cell Signaling Technology (Danvers, MA, USA). Secondary antibodies against rabbit and goat immunoglobulin G (IgG) were obtained from Life Technologies (Carlsbad, CA, USA) and Santa Cruz Biotechnology, respectively. Hemin was purchased from ICN Biomedicals (Irvine, CA, USA). Hemin stock solution (100×) was made by dissolving hemin in 20 mM NaOH.

### 4.2. Cell Culture and Treatment

U373MG cells were obtained from European Collection of Authenticated Cell Cultures (Salisbury, UK). HeLa cells were purchased from ATCC (Manassas, VA, USA). U373MG and HeLa cells were cultured in high glucose DMEM (Sigma-Aldrich, St. Louis, MO, USA) supplemented with 10% FBS and 1x penicillin/streptomycin (Life Technologies, Carlsbad, CA, USA). NHAs were obtained from Lonza (Walkersville, MD, USA) and cultured in astrocyte growth medium (Lonza, Walkersville, MD, USA). Isogenic eIF2α(S/S) and eIF2α(A/A) MEFs were generously given by Randal J. Kaufman (Sanford Burnham Prebys Medical Discovery Research Institute, La Jolla, CA, USA) [24]. Mutant MEF lines lacking four eIF2α kinases (4KO) and their revertants (reconstituted by Hri, Pkr, Perk, or Gcn2) were established by S. Taniuchi and S. Oyadomari [46]. MEFs were maintained in DMEM with 10% FBS plus essential and nonessential amino acids and 1x penicillin/streptomycin (Life Technologies, Carlsbad, CA, USA). Cells were maintained at 37°C in a 5% CO_2_ incubator. For CA treatment, passaged cells were incubated overnight, then the medium was replaced with DMEM supplemented with 3% FBS. After 24 h incubation, the cells were administered with either vehicle (0.1% DMSO), or the appropriate concentration of CA.

### 4.3. Microarray Analysis

U373MG cells were passaged at a concentration of 1.0 × 10^6^ cells/10 cm dish and incubated overnight. The next day, media was changed to fresh DMEM/3% FBS. After 24 h incubation, the cells were administered with either DMSO or 50 μM of CA for 24 h. Total RNAs were prepared using an RNeasy RNA extraction kit according to the manufacturer’s protocol (Qiagen, Hilden, Germany). Microarray analyses were performed using the 3D-Gene Human Oligo chip 25k microarray platform (TORAY, Kamakura, Japan). The fold induction of each gene expression was calculated using Excel software (Microsoft, Redmond, WA, USA). The heat map image was generated by using the web-based program Matrix2png (https://matrix2png.msl.ubc.ca). The microarray data was deposited in the Gene Expression Omnibus (GEO) database (Accession GSE129001).

### 4.4. Quantitative Reverse Transcription PCR (RT-qPCR)

Total RNAs were extracted using TRIzol reagent (Life Technologies, Carlsbad, CA, USA) and reverse transcribed using a PrimeScript II RT kit (Takara Bio, Otsu, Japan) to obtain cDNA samples. qPCR analysis was performed by using SYBR Premix EX Taq II (Takara Bio, Otsu, Japan) and a CFX96 thermal cycler (Bio-Rad, Hercules, CA, USA) as previously described [19]. The sequence of primers used is listed in Appendix A.

### 4.5. Preparation of Cell Lysate

The cytosolic or nuclear extracts were prepared as previously described [19]. Briefly, DMSO- or CA-treated cells were dissolved in hypotonic buffer (10 mM Hepes-KOH (pH 7.9)/10 mM KCl/0.1 mM EDTA) supplemented with a protease inhibitor cocktail and PhosSTOP (Roche, Mannheim, Germany), and NP-40 (Wako, Osaka, Japan) was then added to a final concentration of 0.6%. After centrifugation, the supernatants were stored as the cytosolic fraction, and the separated nuclear pellets were used as the nuclear fraction preparation. To prepare whole cell lysate, the cells were then directly lysed with RIPA buffer (50 mM Tris-HCl (pH 8.0)/150 mM NaCl/1% NP-40/0.5% Na-DOC/0.1% SDS) supplemented with protease inhibitor cocktail and PhosSTOP (Roche, Basel, Switzerland). The protein concentration was determined using a BCA Protein Assay Kit (Thermo Fisher Scientific, Waltham, MA, USA).

### 4.6. Immunoblot

The protein samples were boiled in Laemmli buffer and electrophoresed under reducing conditions on 8% or 10% SDS-PAGE. Proteins were transferred onto polyvinylidene difluoride (PVDF) membranes (Millipore, Billerica, MA, USA) and blocked with 1% skim milk (Wako, Osaka, Japan) or 1% BSA/PBST (PBS(-)/0.1% Tween20) solutions. Primary antibodies against ATF4, Nrf2, lamin B, eIF2α, phospho-Ser51 eIF2α, or HSP90α/β were diluted 1:1000 in 1% BSA/PBST and incubated overnight at 4°C. The primary antibody against PKR was diluted 1:200 in 1% BSA/PBST. After washing with PBST, the membrane was treated with HRP-labeled secondary antibodies (1:5000) against rabbit or goat IgG for 1 h at room temperature. The immunoreactive bands were visualized using ImmunoStar chemiluminescent reagent (Wako, Osaka, Japan).

### 4.7. RNA Interference

The small interfering RNA (siRNA) duplex oligonucleotides targeting human ATF4, Nrf2, HRI, PKR, PERK, and GCN2 were synthesized against the following target sequences: hATF4, 5′-GCC TAG GTC TCT TAG ATG A-3′; hNrf2, 5′-GAG TAT GAG CTG GAA AAA C-3′; hHRI#1, 5′-GAA GTT CTA ACA GGT TTA A-3′; hHRI#2, 5′-GGA CCA ACA GAA ACG GGA A-3′; hPKR, 5′-CGT TGC TTA TGA ATG GTC T-3′; hPERK, 5′-TGG ACC ATG AGG ACA TCA G-3′; hGCN2, 5′-CAG CAG AAA TCA TGT ACG A-3′. A negative control siRNA duplex (cat. #1027310, Qiagen, Hilden, Germany) was used for negative control experiments. For siRNA transfection, 1 × 10^5^ cells were seeded in 12-well plates. The following day, the cells were transfected with 30 nM negative control or gene-specific siRNAs using Lipofectamine RNAiMax Transfection Reagent (Life Technologies, Carlsbad, CA, USA). The medium was replaced with fresh DMEM containing 3% FBS 4 h after transfection. After 20 h incubation, the transfected cells were administered either DMSO or 50 μM of CA for 6 h or 24 h prior to immunoblot or RT-qPCR analysis.

### 4.8. Transient Transfection and Luciferase Assay

To construct pAARE-SV and pARE-SV reporter plasmids, oligonucleotide duplexes containing four AAREs of the human *ASNS* gene (sense, 5′-GAT CTA AGT TTC ATC ATG CCT AAG TTT CAT CAT GCC TGG ATC TAA GTT TCA TCA TGC CTA AGT TTC ATC ATG CCT G-3′; antisense, 5′-GAT CCA GGC ATG ATG AAA CTT AGG CAT GAT GAA ACT TAG ATC CAG GCA TGA TGA AAC TTA GGC ATG ATG AAA CTT A -3′; AAREs are underlined) and four AREs of the mouse *GSTA1* gene (sense, 5′-GAT CTA ATG TGA CAA AGC AAC TTA ATG TGA CAA AGC AAC TGG ATC TAA TGT GAC AAA GCA ACT TAA TGT GAC AAA GCA ACT G-3′; antisense, 5′-GAT CCA GTT GCT TTG TCA CAT TAA GTT GCT TTG TCA CAT TAG ATC CAG TTG CTT TGT CAC ATT AAG TTG CTT TGT CAC ATT A-3′; AREs are underlined) were subcloned into the BglII site of the pGL3 promoter vector (Promega). The nucleotide sequences of all constructs were confirmed by sequencing. For transient transfection, HeLa cells were seeded in 24-well plates 1 day before transfection at a density of 3 × 10^4^ cells/well. The next day, the medium was refreshed with DMEM/3% FBS, and cells were transfected with 400 ng plasmid DNAs (200 ng reporter plasmid and 200 ng co-reporter plasmid (pRL-TK)) using FuGENE HD transfection reagent (Promega, Madison, WI, USA). After 4 h of transfection, medium was changed and DMSO or 50 μM of CA was added. After 20 h incubation, cells were harvested and subjected to a luciferase assay. Luciferase activity was measured using Dual-Luciferase Assay Reagent (Promega).

### 4.9. ChIP Assay

The ChIP assay was performed as previously described [19]. Briefly, U373MG cells were passaged at a density of 1 × 10^6^ cells/10 cm dish and incubated overnight. The next day, the media was refreshed with DMEM/3% FBS. After 24 h incubation, cells were treated with either DMSO or 50 μM of CA for 6 h, fixed with 1% formaldehyde for 10 min at 37°C and lysed with ChIP lysis buffer. The cell lysate was sonicated and subjected to immunoprecipitation with normal rabbit IgG or anti-ATF4 antibody (Santa Cruz Biotechnology, Dallas, TX, USA). The immunoprecipitated DNA was purified and subjected to qPCR analysis. The primer sequences used are as follows: *hASNS*-AARE, 5′-TGG TTG GTC CTC GCA GGC AT-3′ and 5′-CGC TTA TAC CGA CCT GGC TCC T-3′.

### 4.10. Metabolic Labeling

HeLa cells were seeded in 6 cm dishes at a density of 4 × 10^5^ cells/dish and incubated overnight. The next day, media was replaced with fresh DMEM/3% FBS and transfected with either control or anti-HRI siRNA. Following overnight incubation, media was replaced with methionine-free DMEM supplemented with 3% dialyzed FBS (Life Technologies, Carlsbad, CA, USA) for 30 min, and DMSO or CA (final concentration 50 μM) was then added to the culture. After a 30-min incubation, 50 μM L-azidohomoalanine (AHA, Life Technologies, Carlsbad, CA, USA) was added to the cells to label nascent proteins. Following 3 h of additional incubation, cells were washed with PBS once and lysed in lysis buffer (1% SDS/Tris (pH 8.0)) containing protease inhibitors to yield cell lysates. The protein concentration of the cell lysate was measured using a BCA Protein Assay Kit (Thermo Fisher Scientific). Fifty microgram protein samples containing AHA-labeled protein were biotinylated using biotin alkyne and a Click-iT Protein Reaction Buffer Kit according to the manufacturer’s protocol (Life Technologies). The labeled protein was then subjected to SDS-PAGE and transferred onto a PVDF membrane (Pall Corporation, Port Washington, NY, USA). To confirm equal loading, the PVDF membrane was stained with a MemCode Reversible Protein Stain Kit (Life Technologies). After removal of the MemCode stain, the membrane was blotted with Streptavidin-Biotinylated HRP Complex (GE Healthcare, Milwaukee, WI, USA) and visualized with ImmunoStar reagent (Wako Pure Chemicals, Osaka, Japan).

### 4.11. Pull-Down Assay

U373MG cells were treated with DMSO or 50 μM of biotinylated CA (Bio-CA) for 6 h, and then cells were treated with Radio-Immunoprecipitation Assay (RIPA) buffer for whole cell lysate preparation. The whole cell lysate was then incubated with Streptavidin Sepharose Beads (GE Healthcare, Milwaukee, WI, USA) according to the manufacturer’s protocol in order to obtain Bio-CA-bound proteins, which were then subjected to SDS-PAGE and blotted with anti-Keap1 or anti-HRI antibodies.

### 4.12. Statistical and Quantitative Analysis

The Student’s *t*-test or one-way ANOVA with post hoc Tukey–Kramer test was used to estimate the statistical significance of differences between two or more groups. Differences between groups were considered statistically significant with *p* values < 0.05. All experiments were repeated at least three times, and the data are expressed as the mean ± SE (standard error) from representative experiments.

## Figures and Tables

**Figure 1 ijms-20-01706-f001:**
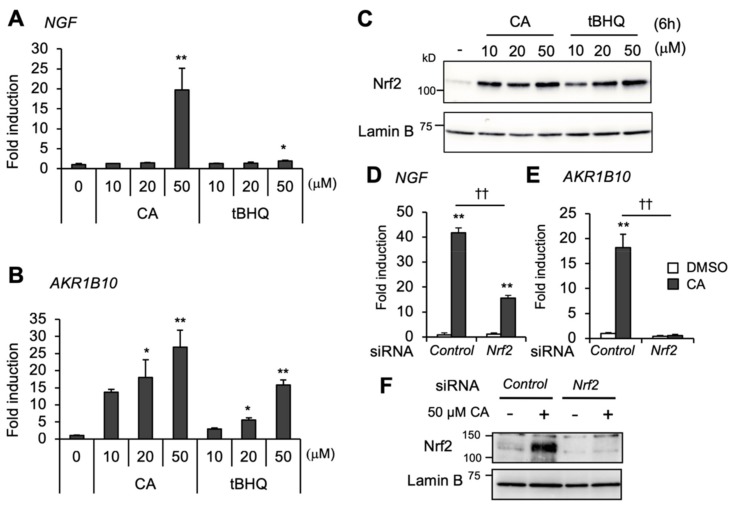
High-dose CA regulates *NGF* gene expression beyond Nrf2 activation. CA and tBHQ induction of (**A**) *NGF* and (**B**) *AKR1B10* genes in U373MG cells. Fold gene inductions were evaluated by reverse transcription-quantitative polymerase chain reaction (RT-qPCR) (vehicle control dimethyl sulfoxide (DMSO) = 1). The results are presented as the mean ± SE (*n* = 3); significant differences from the vehicle control (DMSO) are indicated by asterisks (* *p* < 0.05; ** *p* < 0.01), by one-way analysis of variance (ANOVA) with Tukey–Kramer post hoc test (*n* = 3). (**C**) Nuclear accumulation of Nrf2 protein in response to varying doses of CA or tBHQ was evaluated by immunoblot analysis. Lamin B blot was used as a loading control. (**D**) Effect of Nrf2 knockdown on the CA induction of *NGF* and (**E**) *AKR1B10*. siRNA-transfected cells were treated with either DMSO (open bars) or 50 μM of CA (closed bars) for 24 h. Fold gene inductions were evaluated by RT-qPCR (control siRNA/DMSO = 1). The values are normalized with cyclophilin A expression and presented as the mean ± SE of three independent experiments. Significant differences between DMSO and CA are indicated by asterisks (** *p* < 0.01), and daggers indicate significant differences between control siRNA or Nrf2 siRNA with CA (†† *p* < 0.01) by one-way ANOVA with Tukey–Kramer post hoc test (*n* = 3). (**F**) Nrf2 protein level in Nrf2 knockdown cells. Either control (Ctrl) or Nrf2 siRNA-transfected cells were treated with DMSO or 50 μM of CA for 6 h, then nuclear Nrf2 protein levels were visualized by immunoblot analysis.

**Figure 2 ijms-20-01706-f002:**
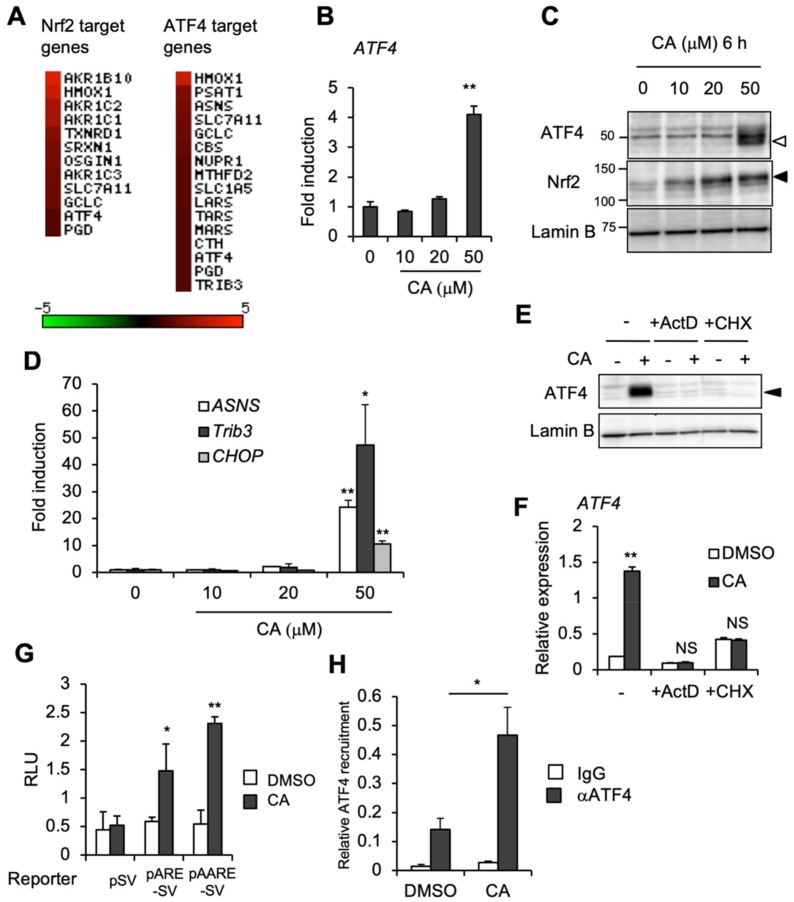
CA activates the ATF4 pathway in a dose-dependent manner in U373MG cells. (**A**) Heat map representation of genes induced by 50 μM of CA in U373MG cells. The fold induction (CA vs. DMSO) was plotted based on microarray data. (**B**) Dose-dependent induction of *ATF4* mRNA. *ATF4* mRNA expression was evaluated by RT-qPCR and presented as the mean ± SE from three independent experiments. Each value was normalized to cyclophilin A expression. The asterisks indicate a significant increase compared with the 0 h control (** *p* < 0.01 by one-way ANOVA with Tukey–Kramer post hoc test). (**C**) CA dose-dependent induction of ATF4 and Nrf2 proteins. U373MG cells were administered 10 to 50 μM of CA for 6 h. An aliquot of nuclear extract was subjected to immunoblot analysis. White and black triangles indicate ATF4 and Nrf2 bands, respectively (**D**) CA dose-dependent induction of ATF4 target gene; mRNA expression of *ASNS*, *Trib3*, and *CHOP* was evaluated as described in (B). The asterisks indicate a significant increase compared with the control (0 μM) (* *p* < 0.05; ** *p* < 0.01 by one-way ANOVA with Tukey–Kramer post hoc test). (**E**) U373MG cells were pretreated with DMSO (-), 1 μg/mL actinomycin D (ActD) or 1 μg/mL cycloheximide (CHX) for 30 min, followed by DMSO or 50 μM of CA for 6 h for immunoblot analysis or (**F**) 24 h for RT-qPCR analysis. Black triangle indicates ATF4 band. The asterisks indicate a significant difference between DMSO- vs. CA-treated samples (** *p* < 0.01; NS not significant by one-way ANOVA with Tukey–Kramer post hoc test). (**G**) AARE-luc reporter gene activation by CA. U373MG cells were transfected with either pSV-Luc (pSV), pARE-SV-Luc (pARE-SV), or pAARE-SV-Luc (pAARE-SV) reporter vector and then incubated for 24 h in the presence of DMSO (open bars) or 50 μM of CA (closed bars). The firefly luciferase activities were normalized to Renilla luciferase activities and presented as relative luciferase units. The values are presented as the mean ± SE. The asterisks indicate a significant increase compared with the DMSO control (* *p* < 0.05; ** *p* < 0.01 by Student’s *t*-test (*n* = 3)). (**H**) CA-dependent ATF4 accumulation with the AARE sequence of human *ASNS* gene. U373MG cells were treated with either DMSO or 50 μM of CA for 6 h and then subjected to ChIP analysis using control IgG (IgG) (open bars) and anti-ATF4 antibody (αATF4) (closed bars). The relative accumulation levels of ATF4 to the *ASNS* gene AARE are presented as the mean ± SE. The asterisks indicate a significant increase compared with the DMSO control (* *p* < 0.05 by Student’s *t*-test).

**Figure 3 ijms-20-01706-f003:**
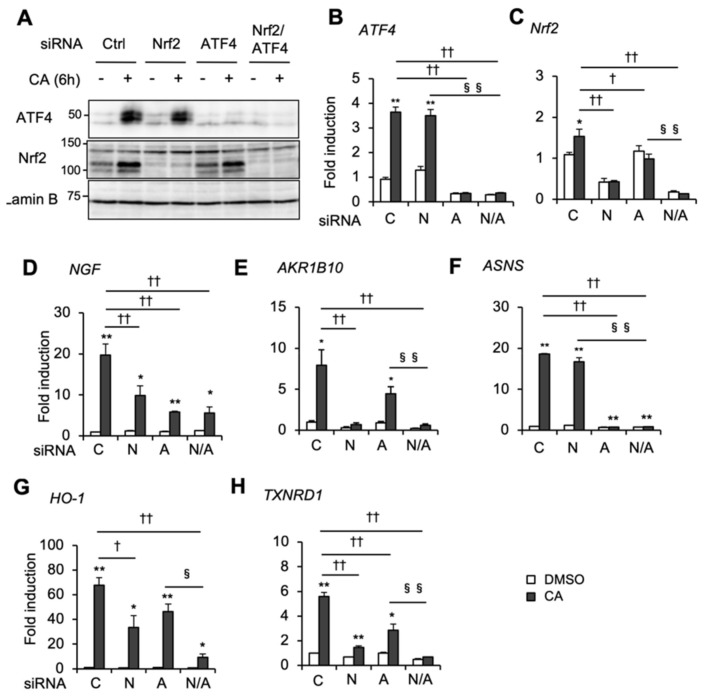
Nrf2 and ATF4 cooperatively modulate CA induction of *NGF* and antioxidant genes. (**A**) ATF4 and Nrf2 protein levels in ATF4 and/or Nrf2 knockdown cells. U373MG cells were transfected with control or gene-specific small interfering RNAs (siRNAs) as indicated. After 24 h transfection, cells were treated with DMSO (-) or 50 μM of CA (+) for 6 h, and an aliquot of nuclear fraction was subjected to immunoblot analysis. (**B**) CA induction of *ATF4*, (**C**) *Nrf2*, (**D**) *NGF*, (**E**) *AKR1B10*, (**F**) *ASNS*, (**G**) *HO-1*, and (**H**) *TXNRD1* genes in siRNA-transfected U373MG cells was evaluated by RT-qPCR analysis (C: control siRNA, N: Nrf2 siRNA, A: ATF4 siRNA, N/A: Nrf2 and ATF4 siRNA). After 24 h siRNA transfection, cells were treated with DMSO (open bars) or 50 μM of CA (closed bars) for 24 h. Each value was normalized to cyclophilin A expression and presented as the mean ± SE from three independent experiments. The asterisks indicate significant differences between DMSO and CA (* *p* < 0.05; ** *p* < 0.01), and the daggers indicate significant decreases from control siRNA with CA vs. Nrf2 and/or ATF4 siRNA with CA († *p* < 0.05; †† *p* < 0.01), and section signs indicate significant differences from Nrf2/ATF4 double knockdown cells with CA (§ *p* <0.05, §§ *p* < 0.01) by one-way ANOVA with Tukey–Kramer post hoc test (*n* = 3).

**Figure 4 ijms-20-01706-f004:**
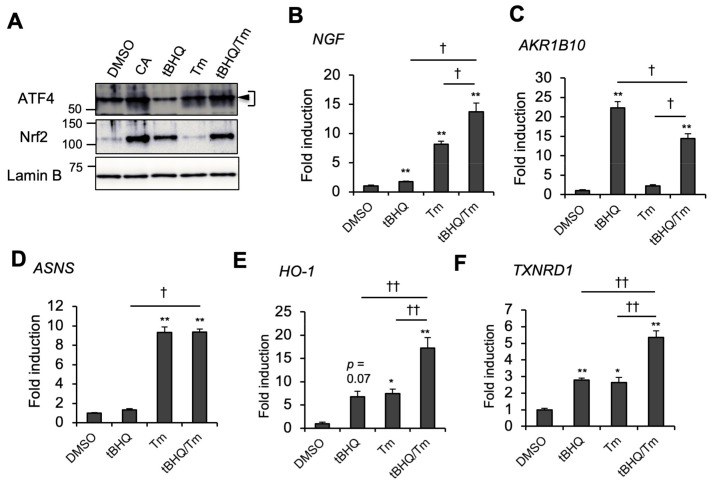
tBHQ and Tm cooperatively modulate *NGF* and antioxidant gene expression. (**A**) ATF4 and Nrf2 protein induction by tBHQ and/or Tm. U373MG cells were treated with DMSO or 50 μM of tBHQ and/or 2 μg/mL of Tm for 6 h, and an aliquot of the nuclear fraction was then subjected to immunoblot analysis. Black triangle indicates nonspecific band in smeared ATF4 band. (**B**) Gene expression of *NGF*, (**C**) *AKR1B10*, (**D**) *ASNS*, (**E**) *HO-1*, and (**F**) *TXNRD1* in response to tBHQ and Tm was evaluated by RT-qPCR analysis. U373MG cells were treated with tBHQ and/or Tm for 24 h. These values were normalized to cyclophilin A expression and presented as the mean ± SE from three independent experiments. The asterisks indicate significant differences from DMSO (* *p* < 0.05; ** *p* < 0.01), and the daggers indicate significant difference between tBHQ or Tm and tBHQ/Tm († *p* < 0.05; †† *p* < 0.01) by one-way ANOVA with Tukey–Kramer post hoc test (*n* = 3).

**Figure 5 ijms-20-01706-f005:**
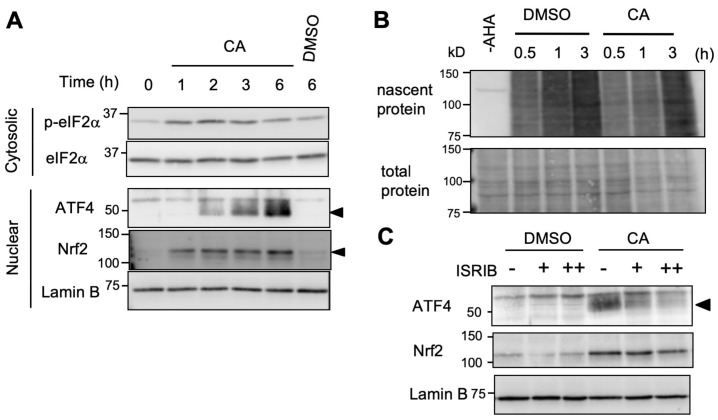
CA activates the ISR pathway for ATF4 induction. (**A**) CA induction of eIF2α serine 51 phosphorylation. U373MG cells were untreated or treated with 50 μM of CA for 1 to 6 h, or DMSO for 6 h, then cytosolic and nuclear fractions were separated and subjected to immunoblot analysis. Phosphorylated eIF2α was detected by anti-phospho-eIF2α antibody. Black triangles indicate ATF4 or Nrf2 band. (**B**) Global translation inhibition by CA. HeLa cells were treated with DMSO or 50 μM of CA in the presence of L-azidohomoalanine (AHA). After 0.5 to 3 h incubation, whole cell extract was prepared and followed by a subsequent Click-iT reaction to label nascent proteins by biotin. Samples were subjected to sodium dodecyl sulfate-polyacrylamide gel electrophoresis (SDS-PAGE) and transferred onto PVDF membrane followed by staining with streptavidin/biotin-horseradish peroxidase (HRP) complex to visualize nascent proteins (upper panel). The same blot was stained with MemCode reagent to visualize total protein (lower panel). (**C**) U373MG cells were pretreated with ISR inhibitor (ISRIB) (- without ISRIB, + 200 nM; ++ 1 μM) for 30 min and then administrated to DMSO or 50 μM of CA. After 6 h incubation, nuclear extracts were subjected to immunoblot analysis. Black triangle indicates ATF4 band.

**Figure 6 ijms-20-01706-f006:**
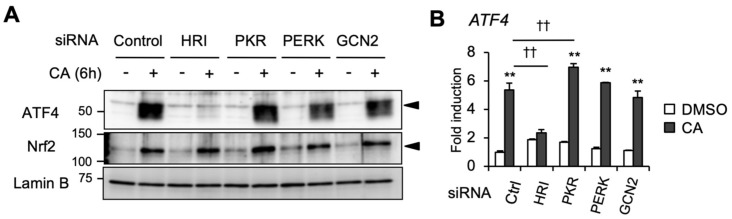
CA activates the ISR pathway through HRI activation. (**A**) The effect of HRI, PKR, PERK, and GCN2 knockdown on CA induction of ATF4 protein. U373MG cells were transfected with either control siRNA or siRNAs targeting HRI, PKR, PERK, and GCN2. After 24 h transfection, the cells were treated with DMSO or 50 μM of CA for 6 h. Nuclear fractions were subjected to immunoblot analysis. (**B**) The effect of HRI, PKR, PERK, and GCN2 knockdown on CA induction of ATF4 mRNA. siRNA-transfected U373MG cells were treated with DMSO or 50 μM CA for 24 h, and *ATF4* mRNA expression was then evaluated by RT-qPCR analysis. The values are presented as the mean ± SE of three independent experiments. The asterisks indicate significant differences between DMSO- and CA-treated samples (** *p* < 0.01), and the daggers indicate significant decreases from control siRNA with CA vs. eIF2α kinase siRNA with CA (†† *p* < 0.01) by one-way ANOVA with Tukey–Kramer post hoc test.

**Figure 7 ijms-20-01706-f007:**
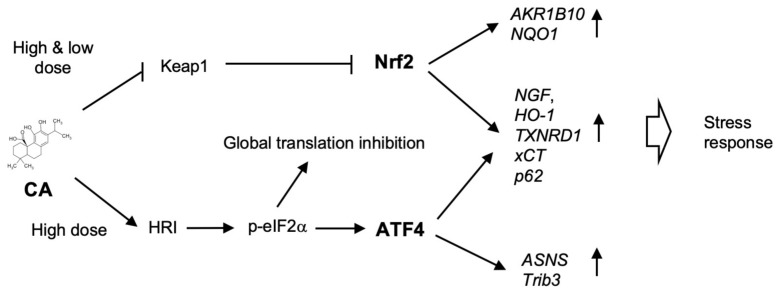
Hypothetical scheme of CA action in mammalian cells. CA activates the Nrf2 pathway only at a low dose (≤20 μM) but activates both Nrf2 and ATF4 pathways at a high-dose (≥50 μM). Induced Nrf2 and ATF4 independently and cooperatively up-regulate CA-inducible gene expression (arrows pointing up) to exhibit physiological effects, such as stress response (bold white arrow). The arrows indicate activation and T-arrows show inhibition.

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
