# Peer review of "Concomitant Nrf2- and ATF4-Activation by Carnosic Acid Cooperatively Induces Expression of Cytoprotective Genes"

_ijms, 2019, doi:10.3390/ijms20071706_

Reviewer 1 Report

Dear Authors, please see the comments below.  "Conceptual" comments are in bold.

Everywhere :  write "50 µM of CA" in stead of "50 µM CA"

Give an outline of the" Integrated stress response (ISR) pathway" early in the introduction

and recapture it in the discussion. 

In the abstract, give the doses of CA in numbers, and what they do .

Line 58 "or" change to "and"

Say in the introduction which cells are used for the study.  Also, mention the doses of CA used (instead of "increasing concentrations" that is at line 74 )

Line 79. Delete "our"

Line 94, in stead of " by unknown factor "  write " by unknown factor or factors "

In the reference of Figure 1 "daggers" should not be mentioned for panels A and B, because there are none.

Line 125, also mention what happens with 20 and 50 µM. 

Line 126 ASN, Trib3, and CHOP please give some explanation what they are, and why they are mentioned, and whether the results are significant.

Line 128  delete "both"

Line 129 in stead of "and mRNA" write "as well as mRNA"

Line 138 In stead of "As" write "Because"

Line 139 in stead of "this" write "the present"

Line 140 say precisely which cells were used (while glioblastoma cells typically are of astrocytic origin, they are no longer astrocytes [not normal astrocytes] you know that, but the reader needs to know which cells were actually used for the study.)

Panel Figure 4D indicate significance or make clear that it is not significant.

Panel Figure 4F, maybe you want to say that in the experimental groups there is absolute no difference between treatment and vehicle, and give some thoughts on it.

Line 171 "severely" is not a good word, try something like "very distinctly" sorry, English doesn't have a good word for this.  All kinds of words that are close, but not precise enough.

Take your pick Synonyms of remarkable

arresting, bodacious, bold, brilliant, catchy, commanding, conspicuous, dramatic,

emphatic, eye-catching, flamboyant, grabby, kenspeckle [chiefly Scottish], marked, noisy,noticeable, prominent, pronounced, showy, splashy, 

striking

The bold ones I do like (-_-)  kenspeckly is good, nobody will understand what you mean. Sorry, I should be a serious reviewer.

Line 176 write "antioxidant genes of the Nrf2/ARE pathway"

Line 177 "treatment with" is not needed.

Line 181 delete "either" in stead of "or" write "as well as" .   In stead of "and" write "while"

Line 193 In stead of "and" write "while"

Line 196 Question :  Are more studies on NHAs planned ?

In figure 3.  Check the "daggers" in panel B, they do not look correct.

Also in A use "C, N, A, N/A" As it stands now, the legend is not correct.

Line  214. Nrf2 and ATF do not appear to match what is presented in the figure.

Line 225.   As said, the ISR should be presented in more detail.Line 228.   "next" can be deleted

Line 231.   Why is it necessary to say "nascent" ?

Figure 5.  the legend does not (fully) cover what is presented in the figure.

Line 253 replace "is" by "may be"

Figure 6. As said, the ISR pathway has not been described / explained. 

It would be good to give quantifications (bar graphs, statistical analysis of panels C-G)

In the legend, give the names GCN2, PERK, HRI, PKR.  In the beginning indicate that panels A and B are described.

Line 276 "then" is not needed

Line 297 "via different molecular mechanism" should by separately and more extensively described.  (in several sentences)

Line 299.  Figure 7 should be used for the discussion.  Meaning figure 7 provides the framework whereby the discussion should be organized.  It is very interesting that the different functional effects of the different concentrations of CA can be explained at least partly by their different effect on, for example, Nrf2 and ATF4.  Perhaps the authors can give examples of other pathways that can be differentially influenced in this way. (low concentrations affecting one molecule, high concentrations affecting additional molecules).

Further, regarding the discussion, the supplemental data should be far less prominent.  Far less detail should be described. Better is to say whether the results are sound, whether they are in accord with the predictions, whether concepts are supported, whether new ideas have been generated.

Line 449 write "vehicle" in stead of "a vehicle"

 Further, the methods appear adequately described.

Author Response

Response to Reviewer 1 Comments

We appreciate the reviewer for your constructive and helpful comments concerning our manuscript. We have revised our manuscript in response to your suggestions. We hope that this revised manuscript is acceptable for publication in IJMS.

Reviewer comment:Everywhere :  write "50 µM of CA" in stead of "50 µM CA"  

Authors Response: We have changed all "50 µM of CA" to "50 µM CA" as suggested.

Reviewer comment:Give an outline of the" Integrated stress response (ISR) pathway" early in the introduction and recapture it in the discussion.

Authors Response: We agree with review's comments, and have included the description about ISR in introduction and recaptured in the discussion with reference. (Line 68-76).

Reviewer comment:In the abstract, give the doses of CA in numbers, and what they do . 

Authors Response: We have given CA doses and added sentence as suggested (Line29-30).

Reviewer comment: Line 58 "or" change to "and" 

Authors Response: We have changed "or " to "and" (Line 60).

Reviewer comment:Say in the introduction which cells are used for the study.  Also, mention the doses of CA used (instead of "increasing concentrations" that is at line 74 ) 

Authors Response: We have described cell line name and concentration of CA used (Line 82-84).

Reviewer comment:Line 79. Delete "our" 

Authors Response: We have deleted (Line 89).

Reviewer comment:Line 94, in stead of " by unknown factor " write " by unknown factor or factors " 

Authors Response: We have changed (Line 104).

Reviewer comment:In the reference of Figure 1 "daggers" should not be mentioned for panels A and B, because there are none. 

Authors Response: Thank you for pointing out the error. We have removed (Line 110).

Reviewer comment:Line 125, also mention what happens with 20 and 50 µM. 

Authors Response: We have added sentence in the manuscript (Line 137).

Reviewer comment:Line 126 ASN, Trib3, and CHOP please give some explanation what they are, and why they are mentioned, and whether the results are significant. 

Authors Response: We have added the explanation about ASNS, Trib3, and CHOP in the manuscript (Line 138-139).

Reviewer comment:Line 128  delete "both" 

Authors Response: We have deleted "both" (Line 141).

Reviewer comment:Line 129 in stead of "and mRNA" write "as well as mRNA" 

Authors Response: We have changed (Line 141).

Reviewer comment:Line 138 In stead of "As" write "Because" 

Authors Response: We have changed (Line 152).

Reviewer comment:Line 139 in stead of "this" write "the present" 

Authors Response: We have changed (Line 153).

Reviewer comment:Line 140 say precisely which cells were used (while glioblastoma cells typically are of astrocytic origin, they are no longer astrocytes [not normal astrocytes] you know that, but the reader needs to know which cells were actually used for the study.) 

Authors Response: We agree with this comment. "astrocytic cells" is ambiguous description, and we have given actual name of cell line (U373MG) (Line 154).

Reviewer comment:Panel Figure 2D indicate significance or make clear that it is not significant. 

Authors Response: Thank you for pointing out the oversight. We have added asterisks to indicate significance in Figure 2D, and added the definition in figure legend (Line 165-167).

Reviewer comment:Panel Figure 2F, maybe you want to say that in the experimental groups there is absolute no difference between treatment and vehicle, and give some thoughts on it. 

Authors Response: We agree to this comment. What we want to say here is that Act D or CHX pretreatment abolished CA induction of ATF4. Thus, we have revised figure as revised Figure 2F, and changed the figure legend (Line 170-171).

Reviewer comment:Line 171 "severely" is not a good word, try something like "very distinctly" sorry, English doesn't have a good word for this. All kinds of words that are close, but not precise enough. 

Authors Response: We have changed "severely" to "almost totally" (Line 188).

Reviewer comment: Line 176 write "antioxidant genes of the Nrf2/ARE pathway" 

Authors Response: Thank you for the kind comment. We have changed as suggested (Line 193).

Reviewer comment:Line 177 "treatment with" is not needed. 

Authors Response: We have deleted "treatment with" (Line 194-195).

Reviewer comment:Line 181 delete "either" in stead of "or" write "as well as". In stead of "and" write "while" 

Authors Response: We have revised as suggested (Line 198-199).

Reviewer comment: Line 193 In stead of "and" write "while" 

Authors Response: We have changed "and" to "while" (Line 210).

Reviewer comment:Line 196 Question: Are more studies on NHAs planned ?

Authors Response: Thank you for the comment. Currently, we have no plan to perform additional experiments.

Reviewer comment: In figure 3. Check the "daggers" in panel B, they do not look correct. 

Authors Response: Thank you for pointing out the oversight. "daggers" in panel B was incorrect, and we have deleted (Figure 3B).

Reviewer comment:Also in A use "C, N, A, N/A" As it stands now, the legend is not correct. 

Authors Response: Thank you for pointing out the oversight. We deleted this sentence (Line 220-221), and inserted in an appropriate position (Line224-225).

Reviewer comment:Line 214. Nrf2 and ATF do not appear to match what is presented in the figure. 

Authors Response: Thank you for pointing out the oversight. We have changed "Nrf2 and ATF4" to "tBHQ and Tm" (Line 233).

Reviewer comment:Line 225. As said, the ISR should be presented in more detail.

Authors Response: We have added the detailed description about ISR in the Introduction section at Line 245-247.

Reviewer comment:Line 228. "next" can be deleted 

Authors Response: We have deleted "next" (Line 248).

Reviewer comment:Line 231. Why is it necessary to say "nascent" ? 

Authors Response: We agree this comment, and have deleted "nascent" (Line 251).

Reviewer comment:Figure 5. the legend does not (fully) cover what is presented in the figure.

Authors Response: Thank you for pointing out the oversight. we have revised Figure 5 legend (Line261-272).

Reviewer comment:Line 253 replace "is" by "may be"

Authors Response: We have changed "is" to "may be" (Line 274).

Reviewer comment:Figure 6. As said, the ISR pathway has not been described / explained. 

Authors Response: Please see the response above.

Reviewer comment: It would be good to give quantifications (bar graphs, statistical analysis of panels C-G) 

Authors Response: We thank the reviewer for this comment. We think the reviewer is right, such quantifications are useful to understand the mechanism of HRI activation by CA, and we will do it in the future.

Reviewer comment:In the legend, give the names GCN2, PERK, HRI, PKR. In the beginning indicate that panels A and B are described. 

Authors Response: We have given "HRI, PKR, PERK, and GCN2" in figure legend in the beginning of the description of (A) and (B) (Line 301-306).

Reviewer comment:Line 276 "then" is not needed 

Authors Response: We have deleted (Line 307).

Reviewer comment:Line 297 "via different molecular mechanism" should by separately and more extensively described. (in several sentences) 

Authors Response: We agree this comment, and revised the description about mechanism (Line 327-331).

Reviewer comment:Line 299. Figure 7 should be used for the discussion. Meaning figure 7 provides the framework whereby the discussion should be organized. It is very interesting that the different functional effects of the different concentrations of CA can be explained at least partly by their different effect on, for example, Nrf2 and ATF4.  Perhaps the authors can give examples of other pathways that can be differentially influenced in this way. (low concentrations affecting one molecule, high concentrations affecting additional molecules). 

Authors Response: We thank the reviewer for this comment. Following reviewer's comments, we have rewritten discussion and give resveratrol as an example of multi-target compound (Line 331-336).

Reviewer comment: Further, regarding the discussion, the supplemental data should be far less prominent. Far less detail should be described. Better is to say whether the results are sound, whether they are in accord with the predictions, whether concepts are supported, whether new ideas have been generated. 

Authors Response: We thank the reviewer for this comment. We have reorganized discussion section to relocate description for supplemental data as much as possible to results section (Line 286-291, Line 295-298).

Reviewer comment:Line 449 write "vehicle" in stead of "a vehicle" 

Authors Response: we have changed (Line 494).

Reviewer comment:Further, the methods appear adequately described.

Authors Response: We thank the reviewer for this encouraging comment.

Reviewer 2 Report

The manuscript by Mimura et al describes work to investigate a novel mechanism of CA-mediated gene regulation evoked by Nrf2 and ATF4 cooperation. The manuscript is mostly well written but there are some needs to be improved.

Line 55-57, We previously found that CA induces nerve growth factor (NGF) in U373MG human astrocytoma cells in an Nrf2-dependent manner [6, 7]... Did you use U373MG human strocytoma cells?  Other cells?

Line 74-75, We found that increasing concentrations of CA activate ATF4 in addition to Nrf2, and ATF4, and Nrf2 cooperatively activate CA induction of antioxidant genes and NGF. You should rewrite.

Author Response

Response to Reviewer 2 Comments

We appreciate the reviewer for your constructive and helpful comments concerning our manuscript. We have revised our manuscript in response to your suggestions. We hope that this revised manuscript is acceptable for publication in IJMS.

Reviewer comment:Line 55-57, We previously found that CA induces nerve growth factor (NGF) in U373MG human astrocytoma cells in an Nrf2-dependent manner [6, 7]. Did you use U373MG human astrocytoma cells? Other cells? 

Authors Response: Thank you for this comment. We have described the name of cell lines (Line 57).

Reviewer comment:Line 74-75, We found that increasing concentrations of CA activate ATF4 in addition to Nrf2, and ATF4, and Nrf2 cooperatively activate CA induction of antioxidant genes and NGF. You should rewrite. 

Authors Response: We have rewritten this sentence. (Line 83-84).

Round  2

Reviewer 1 Report

Reviewer comment: It would be good to give quantifications (bar graphs, statistical analysis of panels C-G) 

Authors Response: We thank the reviewer for this comment. We think the reviewer is right, such quantifications are useful to understand the mechanism of HRI activation by CA, and we will do it in the future.

This being the case, omit this figure from the body of the results.  These results then can be given in the text of the Discussion as preliminary results.  Or as unpublished results. The figure can be given as a supplemental file. 

After having done this, please modify the text of the results and the discussion, accordingly.

Thus, the paper really has been improved considerably, it only needs adequately dealing with this last point.

Author Response

Response to Reviewer 1 Comments

Reviewer comment:It would be good to give quantifications (bar graphs, statistical analysis of panels C-G) 

This being the case, omit this figure from the body of the results.  These results then can be given in the text of the Discussion as preliminary results.  Or as unpublished results. The figure can be given as a supplemental file. 

After having done this, please modify the text of the results and the discussion, accordingly.

Thus, the paper really has been improved considerably, it only needs adequately dealing with this last point.

Authors Response: We appreciate the reviewer for constructive comments, and are sorry for our inappropriate response. 

As suggested, we have removed Figure 6 (C), (D), (E), (F), and (G), and the text (Line 308, and 313) from the results. We have presented these deleted figures as Supplementary Figure S8, and rewritten discussion section (Line 479-485).

Round  3

Reviewer 1 Report

The manuscript looks fine now.